# The Influence of Different Pretreatment Methods on Color and Pigment Change in Beetroot Products

**DOI:** 10.3390/molecules26123683

**Published:** 2021-06-16

**Authors:** Emilia Janiszewska-Turak, Katarzyna Rybak, Ewelina Grzybowska, Ewelina Konopka, Dorota Witrowa-Rajchert

**Affiliations:** Department of Food Engineering and Process Management, Institute of Food Sciences, Warsaw University of Life Sciences–SGGW, 02-787 Warsaw, Poland; katarzyna_rybak@sggw.edu.pl (K.R.); s192910@sggw.edu.pl (E.G.); ewelina.konopka7991@gmail.com (E.K.); dorota_witrowa_rajchert@sggw.edu.pl (D.W.-R.)

**Keywords:** beetroot, ultrasound, steam treatment, color, betalain

## Abstract

Vegetable processing pomace contains valuable substances such as natural colors that can be reused as functional ingredients. Due to a large amount of water, they are an unstable material. The aim of our research was to assess how the pretreatment method (thermal or nonthermal) affects the properties of powders obtained from beet juice and pomace after the freeze-drying process. The raw material was steamed or sonicated for 10 or 15 min, and then squeezed into juice and pomace. Both squeezed products were freeze-dried. The content of dry substance; L*, a*, and b* color parameters; and the content of betalain pigments were analyzed. Pretreatments increased the proportion of red and yellow in the juices. Steam and ultrasound caused a significant reduction in parameter b* in the dried pomace. A significant increase in betanin in lyophilizates was observed after pretreatment with ultrasound and steam for 15 min. As a result of all experiments, dried juices and pomaces can also be used as a colorant source. However, there is higher potential with pomaces due to their additional internal substances as well as better storage properties. After a few hours, juice was sticky and not ready to use.

## 1. Introduction

Pigments are becoming increasingly used in food production. Many synthetics can have a negative effect on the human body. Therefore, natural pigments are playing a large part in food processing applications. They can be obtained, for example, from red pepper, carrots, beetroot, spinach, or grapes.

Beetroot (*Beta vulgaris*) is a plant characterized by its high nutritional value and strong antioxidant properties [1,2]. It contains large amounts of vitamins (vitamin A, thiamine, riboflavin, niacin, pantothenic acid, vitamin B6, ascorbic acid, and folic acid), minerals (sodium, calcium, iron, phosphorus, potassium, magnesium, and zinc), and substances responsible for color (mainly betanin and vulgaxanthin I). The composition of the beetroot may vary depending on the variety [3,4,5].

Most of the beetroot produced is eaten as a vegetable. The remaining part is processed into dried and frozen products, juices and their concentrates, as well as food pigments [6]. Usually, to improve the efficiency of the pressing process, pretreatment of vegetables and fruits is applied. Such methods include, among others, blanching, steaming, and ultrasonic treatment.

Steam treatment consists of heating the raw material in water vapor at a temperature of 100 °C; most often, the process occurs under atmospheric pressure. The steam treatment causes tissue softening, swelling and sticking of starch, and the dissolution of selected nutrients [7]. However, high temperature is a factor causing the loss of many thermolabile components, including betalain pigments. The red-violet betanin breaks down under the influence of heat into cyclo-DOPA and betalamic acid. The reaction can be partially reversed by influencing the pH of the environment, the temperature range, and its duration. Red pigments decompose much faster by heat treatment compared to yellow betaxanthins [3].

In turn, pretreatment with ultrasound (US), i.e., sound waves with a frequency exceeding the threshold of human hearing (from 20 kHz to 1 GHz), is an interaction with the use of vibrational energy generated by an ultrasonic transducer, which converts electrical energy into acoustic energy. Sound waves are divided into high frequency (low energy, low intensity) at frequencies >100 kHz (MHz range) and low frequency (high energy, high intensity) at a frequency range of 20–100 kHz [8,9,10]. During the process of high-energy and high-intensity US, the raw material is compressed and expanded, creating microscopic channels in the structure. These created spaces often lead to the leakage of intracellular fluid into the environment [10,11]. In solids, the impact of ultrasound causes the “sponge effect”. As a result of the propagation of sound waves in the medium, microscopic channels are created, which facilitate the removal of water from the inside of the sonicated material to the external environment. The action of ultrasound is usually created by the impact of these sound waves and is considered a result of cavitation, thermal mechanism, and stress mechanism [10,12].

In the production of juices, large amounts of waste in the form of pomace are generated. Recent studies show that pomace and waste from fruit and vegetable processing contain valuable substances such as antioxidants, dietary fibers, natural pigments, and aromas, which can be reused as functional ingredients [3,13,14]. Beet pomace is rich in bioactive substances and pigments. For this reason, beet pomace can be used to enrich other products with nutrients and color [15]. However, due to its large amount of water, it is perishable. Therefore, increasing its durability through drying would enable its better use [16].

The use of freeze-drying (FD), in most cases without the addition of auxiliary substances, allows obtaining a good quality product using all or part of the raw material that interests us. Moreover, the use of low temperatures in the process allows the maintenance of a significant amount of thermolabile compounds in the material compared to the spray-drying method [17]. Freeze-drying is the removal of water from a previously frozen product owing to the sublimation process. The process is conducted at a very low pressure and low temperature. The product is frozen at a temperature of −20 to −40 °C. The pressure is usually lowered to 0.13–1.3 hPa (usually 0.6–0.7 hPa). Freeze-drying allows for the removal of 80–95% of water contained in the product. The end of this process is the equivalent of reaching a positive temperature in the food. Freeze-drying consists of three stages: freezing, sublimation, and desorption [18].

Usually, freeze-drying produces good-quality food with well-retained flavor and nutritional quality. An effective freeze-drying process can maintain the volume of the material, such as pomace, but the obtained product is usually highly porous, fragile, and hygroscopic [19,20,21]. The porous structure created during the freeze-drying process is useful in the production of food powders because of its good rehydration ability. Nevertheless, with this porous structure, active compounds can be affected by oxygen or water from the environment, while the porosity increases the contact area [17,18].

As previously mentioned, after pressing the juice, pomace is created as a by-product. In the pomace, much pigment is left after pressing. Furthermore, for the drying of juices, usually by spray-drying or the freeze-drying method, additional substances are needed, such as maltodextrin, Arabic gum, etc. [22,23]. Pomace without any additional substances is a more interesting material for future use in the food industry. Therefore, the aim of this study was to test if the obtained pomace could be classified as a pigment source. Moreover, freeze-drying was used to create a powdered form of pomace.

The scope of our work included: (a) pretreatment of red beet, (b) obtaining juice and pomace, and (c) drying the obtained juices and pomace by freeze-drying. At each stage in juices and pomace, the content of dry substance, color, and the content of betalain pigments were determined (a) to verify the influence of the pretreatment methods on color determinants and (b) to compare the juice and pomace colorants and color content. 

## 2. Results and Discussion

Raw beetroot was characterized by dry matter (d.m.) 24.657 ± 1.24%, pigment content 43.5 + 17.9 g/100 g d.m., and 18.3 ± 7.6 g/100 g d.m for betalain and vulgaxanthin-I, respectively. Color coefficients (L* 9.92 + 1.32, a* 23.29 ± 1.96, and b* 6.97 ± 0.74) on the color wheel corresponded with the color purple (Appendix A). The physico-chemical parameters of vegetables depend on soil, weather, and harvesting time. These factors account for the differences in the results for raw beetroot obtained by Fijałkowska et al. [24], and Fijałkowska et al. [25], who measured the same parameters before convective drying. These researchers stated that their analyzed beetroot had a dry matter of 14.3%; color coefficients of L* = 20.3, a* = 21, and b* = 5.6; a red pigment content of about 500 mg/100 g d.m.; and a yellow pigment content of about 400 mg/100 g d.m. [24,25]. Nistor et al. [26] and Seremet Ceclu et al. [1] also mentioned that their analyzed beetroot was characterized by a dry matter of 12–13.74%. Therefore, the beetroot obtained for our investigation had more dry matter than those presented in the literature.

In the next investigation, juices and pomace obtained after the pressing of beetroot are described.

### 2.1. Overall Physical Properties of Juice and Pomace

Generally, the use of pretreatment on vegetable tissues is dictated mainly by the desire to loosen its internal structures, which contributes to a greater extraction of juice and substances closed within the tissue structures during pressing [10,12]. In the present study, the influence of two different types of pretreatment was used. Steam treatment was included in the thermal treatment, while US is a nonthermal technology. 

The amount of dry matter in juices and pomace was different from the raw material but also from each other (Table 1). The lowest amount of dry matter was obtained for juices, independent of pretreatment, and the highest amount was observed for raw material. This situation was related to the pressing process, which caused the removal of water and other soluble substances from the beet tissue. 

A higher dry matter content was observed in pretreated beet juice and pomace, which may be related to the release of water-soluble substances from the tissue. The dry matter content for pretreated beet juice and pomace contrasts with the results for beetroot slices, in which the highest d.m. was observed for the untreated beetroot slices [24,25]. However, the main difference was the tested material; in the research presented, this was juice and pomace (small parts of beetroot).

Used before pressing, different pretreatment methods did not cause any statistically significant changes to dry matter in juices and pomace compared with those obtained without pretreatment (Table 1). For juices, only steam administered for 15 min produced a statistical increase in dry matter content. Similar results were obtained by Fijałkowska et al. [25] and Fijałkowska et al. [24], who used US for 10, 20, and 30 min on the beetroot tissue. No influence on the dry matter was seen after 10 and 20 min.

A different situation was observed after the freeze-drying process of juices. For those juices obtained after the pressing of steamed beetroot, the dry matter after FD was lower than for untreated freeze-dried juices. However, for the US treatment, no statistically significant change was observed. The dry matter of freeze-dried pomace did not differ statistically from the dried untreated pomace. However, the time length analyzed for both pretreatment methods showed an increase in dry matter after a longer period of pretreatment.

Freeze-dried powders of pomace and juices characterized by dry matter higher than 93% (juices 93.7–95.1%; pomace 97.2–97.9%) guarantees the stability of the obtained powders.

The freeze-drying process caused the removal of water from juices and pomace. All obtained dried juices and pomace were stable because of their high dry matter content (Table 1). Dried material can be referred to as powder when its dry matter content is higher than 85–90%. High dry matter or low water content in dry materials ensures microbiological safety by preventing microbial spoilage, lipid oxidation, and quality problems such as agglomeration [27]. In the literature, there is less information about juices and pomaces obtained from pretreated material. Usually, raw vegetables or fruits are tested. Pretreatments, such as US and blanching of beetroot slices, were tested by Ciurzyńska et al. [28].

Lower dry matter in dried juices after pretreatment may be related to the dry matter of juices and pomace before drying. Therefore, with higher d.m. in the raw material, and simultaneously FD, it is possible to obtain a higher d.m. in the dried vegetable.

The juices contained, except for active ingredients, a high amount of low molecular substances such as sugars. These sugars are the reason why, after drying, the powder was sticky and lumped very quickly. This situation was observed during the experiments presented, where, after a few hours of storage, the powders became sticky and aggregated. In this form, they are not accepted for future use as a pigment source. Before the drying process, high molecular substances such as maltodextrin or gum are added, which increase the glass temperature (T_g_) of the powder and help with protection by changing the state of the product during storage [22,29]. 

### 2.2. Color Analysis

The highest value of all color coefficients was observed in juices and pomace obtained from beetroot after 15 min of pretreatment (Table 1). Changes in pomace color coefficients showed no significant influence on lightness (L*) and redness (a*) coefficients after using US pretreatment, whereas for steam usage, a decrease in L*, a*, and b* coefficients was observed, as well as for yellow (b*) using the US treatment (Table 1). For color coefficients of juices, a significant increase in red and yellow was observed. 

Color coefficients, especially the lightness L*, may be related to the water content in the tested material; the higher the water content in the material, the lighter the product. After pretreatment, it may also be related to the release of water-soluble pigments into the water used in both pretreatments. However, it may also be related to the degradation of pigments during pretreatment time [30], or the pressing process in which pigments being squeezed have contact with oxygen and are generated via process heating [31,32].

An increase in the pretreatment time of beetroot did not cause changes in the color coefficient after pressing juices, independent of the pretreatment type. It may be related to the short time of pretreatment. For example, the same correlation was mentioned in the literature for beetroot slice pretreatment using US for 10, 20, and 30 min, and blanched beetroot slices [24,25]. Therefore, after the pressing process, the same correlation occurred in juices and pomace. 

After the freeze-drying of pomace and juices, a significant change was observed (Table 1). Analyzing the changes that appeared after the freeze-drying process, we observed for pomace that increased time of ultrasound treatment caused an increase in red and yellow. In the rest of the samples, no change was seen. More modifications were observed in freeze-dried juices (Appendix A). A decrease in red and yellow was observed by comparing color coefficients of untreated material to those obtained after pretreatment. Moreover, no similar trend was observed by increasing the time of the pretreatment process for the L*, a*, and b* coefficients. Increasing the time of the steam process resulted in FD juices significantly increasing in lightness and yellow coefficient values, but a significant decrease in redness values, whereas for the US treatment, the opposite observation was recorded. The exception was yellow for which no change after increasing the time length was observed. 

The color of food is mainly related to the pigment content, and the color coefficients may be dependent on them. However, betalains from beetroot are divided into two groups, betacyanins and betaxanthins [33,34], which, via higher temperatures, can reach the degradation of Cyclo-DOPA and betalamic acid (the reaction can be reversible) [4,35]. Faster degradation was also observed for red beetroot pigments than for yellow ones [36,37].

### 2.3. Pigment Content Analysis

The highest value of red and yellow pigments was identified in untreated samples of both juice and pomace (Figure 1). The use of pretreatment on the beetroot before pressing, independent of the type of pretreatment, caused a significant decrease in both pigment contents. The increase in pretreatment time caused a significant increase only for red pigment (betalain) in juices after using ultrasound. In other cases, no significant change was observed. 

After freeze-drying, some trend in pigment changes was observable. However, in most cases, it was statistically insignificant. Exceptions included changes observed for betalain pigment (red) in untreated pomace and obtained after 15 min of US.

As mentioned in the literature, betalains are water-soluble pigments, so both pretreatment methods may cause betalains to enter the water in the US baths as well as into steam [22,38,39]. More factors affecting the stability of betalains include: the presence of oxygen, light, temperature, heat treatment, and water activity of the product [40,41,42]. Higher temperature, as in the steam pretreatment, results in the degradation of betalains. Bassama et al. [43] tested the thermal degradation of betacyanin during pasteurization in a temperature range from 60 to 90 °C and they recorded the creation of a brown shade of cactus pear. This betalain degradation was described by first-order reaction kinetics and was positively correlated with temperature. Additionally, a comparison of thermal stability of pigments from food was performed by Gimenez et al. [44], who observed that betacyanin extracts are more temperature-sensitive than anthocyanin or carotenoid extracts.

Furthermore, US applications have a pronounced effect on this colorant. It is mainly connected with better extraction because of the rupturing beetroot cell walls and release of pigments from the plant tissue. This, as well as ultrasonic cavitation, improved the transport of the colorants to the external medium [24,45]. Fijałkowska et al. [24] observed, in higher amounts than in the present study, beetroot pigment in beetroot slices pretreated by US for 20 min. This finding may be related to the tested material type in raw and dried juice and pomace, not the beetroot slices.

To establish a relationship between color coefficients and pigment content, a linear correlation was drawn (Appendix A).

Only the correlation for red pigment in juice and a* coefficient was significant (a* = −0.0081 × red pigment + 15.22, r = 0.958; from the Pearson table, r should be higher than 0.8783). The rest of the correlations were insignificant, so no prediction based on color coefficients can be made without testing the pigment content.

## 3. Materials and Methods

### 3.1. Materials

Beetroot (*Beta vulgaris*) was purchased from a local supermarket in Warsaw, Poland, and stored at a temperature range of 4–6 °C before being used. 

### 3.2. Technological Treatment

#### 3.2.1. Pretreatment

The beetroot was peeled, halved, and chopped into identical parts with the same material conditions. For each pretreatment, 200 g was used and the experiment was conducted in duplicate. After pretreatment, the beetroot was gently dried on filter paper to avoid water leakage. Beetroot without and after pretreatment was used to obtain juice. 

##### Steam Pretreatment

The thermal pretreatment process was conducted in a food steamer, intellisteam model 48,780 (Morphy Richards, U.K.). The raw beetroot samples were placed in the machine and the “vegetable” steaming program was selected for 10 or 15 min.

##### Ultrasound

The beetroot samples were placed in an ultrasonic bath (MKD-3, MKD Ultrasonics, Stary Konik, Poland, internal dimensions: 240 × 140 × 110 mm) at a frequency of 21 kHz, with a total power of sonotrodes of 300 W for 10 and 15 min. The ratio of beetroot to water was 1:4. 

#### 3.2.2. Juice Pressing

The pretreated beetroot and beetroot slices without pretreatment were used to obtain juice. The juice was pressed with a juicer model, NS-621CES (Kuvings, NUC Electronics Co., Ltd., Korea). Juice and pomace were collected separately.

#### 3.2.3. Freeze-Drying

Obtained juices and pomace were placed on a petri dish and frozen at −40 °C (Shock Freezer HCM 51.20, Irinox, Treviso, Italy) for 5 h. Freeze-drying was conducted in an ALPHA 1–4 freeze dryer (Christ, Osterode am Harz, Germany) for 24 h at a shelf temperature of 30 °C. A constant pressure of 63 Pa and a safety pressure of 103 Pa were set. Dried samples were ground (KA A11 basic, IKA Werke GmbH and Co. KG, Staufen, Germany) and stored in glass vessels in a refrigerator until testing.

### 3.3. Analytical Method

#### 3.3.1. Dry Matter 

Dry matter was determined in beetroot as well as in fresh and ground freeze-dried juices and pomace. All samples were measured gravimetrically. About 0.6–1 g of the sample was placed in a dish and dried using the vacuum drying method (Memmert VO400, Schwabach, Germany) under a pressure of 10 mPa in 75 °C for 24 h and constant weight, according to information from Rybak et al. [46] and the AOAC protocol [47]. Juices were placed on dried filter paper placed in a dish, which prevented the juice from burning. Measurements were recorded in triplicate.

#### 3.3.2. Color Analysis 

The color components were measured using a CR-5 colorimeter (Konica Minolta, Japan) in the CIE Lab system. The protocol of measurement was described by Rybak et al. [31]. The color values were recorded as brightness (L*), the proportion of green (negative a* values) or red (positive a* values), and the proportion of blue (negative b* values) or yellow (positive b* values). Calibration was performed with a white pattern (L* 92.49, a* 1.25, b* −1.92). The measurement was recorded on a glass, transparent petri dish onto which the pomace, juice, or powder was placed at the 5 mm layer, with standard illumination C, illuminant D65, and angle for observation of 2°. All measurements were recorded in five repetitions.

#### 3.3.3. Betalain Analysis 

Quantification of betalain was performed by the spectrophotometric method described by Janiszewska and Włodarczyk [48] with some modifications. For this measurement, a Helios Gamma spectrophotometer (Thermo Spectronic, Cambridge, U.K.) was used. Pigments were extracted from the sample with a phosphate buffer with pH 6.5. A sample of 0.7 g (fresh juice, pomace) or 0.05 g (freeze-dried juice, pomace) was mixed for 10 min with 50 mL of phosphate buffer. All measurements were recorded in three repetitions for each sample.

The determination of betalain concentration, i.e., red and yellow pigments, was calculated in terms of betanin (mg betanin/100 g d.m.) and vulgaxanthin-I (mg vulgaxanthin-I/100 g d.m.), respectively. Pigment content calculations were based on the absorption values A1%, which are 1120 for betanin (at 538 nm) and 750 for vulgaxanthin-I (at 476 nm). According to the methodology, the absorbance at 600 nm was measured and used to correct the amount of impurities [45].

For red pigment, calculation Equation (1) was used. This calculation considers the presence of impurities (*A*_600_).
(1)$$Red=\frac{R\cdot 1.095\left(A_{538}-A_{600}\right)}{m\cdot d.m.\cdot 1120}$$

For yellow pigments, calculation Equation (2) was used. This calculation considers the absorbance of red pigments:(2)$$\mathrm{Yellow}=\frac{R\cdot \left(A_{476}-A_{538}+0.677\cdot 1.095\left(A_{538}-A_{600}\right)\right)}{m\cdot d.m.\cdot 750}$$
where *R* is the sample dilution (here, 5000 mg); *A*_476_, *A*_538_, and *A*_600_ are the absorbance of the solution determined at 476, 538, and 600 nm, respectively; *m* is the sample mass (g); *d.m*. is dry matter (*g*/*g*); 1.095 is the coefficient correcting for the increase in absorbance caused by the presence of impurities at 538 nm; 1120 is the absorbance of 1% betanin solution determined at a wavelength of 538 nm in a 1 cm cuvette; 0.677 is the coefficient correcting for the increase in absorbance caused by the presence of impurities at 476 nm; and 750 is the absorbance of 1% vulgaxanthin solution determined at a wavelength of 476 nm in a 1 cm cuvette.

### 3.4. Statistical Treatment

The obtained results were subjected to a statistical analysis using Statistica 13 software (StatSoft, Poland), using one-way analysis of variance with Tukey’s HSD test at a significance level of α = 0.05. The other parameters were determined using MS Excel 16.

## 4. Conclusions

In this research, juice without carrier addition was used, as well as pomace considered production waste. The main goal was to produce pigment from the pomace as well as the juice. The creation of powders from juice and pomace using the freeze-drying process resulted in good-quality powders that can be used as a colorant source. However, powder from juices, which was sticky and lumpy after a few hours of finishing the freeze-drying process, requires packing in our opinion. These unfavorable characteristics are linked to a higher water intake from the environment as well as a low glass transition temperature. With the powders obtained from pomace, this situation was not observed because of the presence of tissues. 

Pretreatment used in this research caused a decrease in pigment content (betalain and vulgaxanthin-I), independent of time and type of pretreatment. However, for 15 min, the US treatment resulted in a minor decrease in pigment content in juices and pomace. Higher pigment content was observed in powders obtained from juices compared with pomace. Pretreatments increased the proportion of the colors red and yellow in the juices. Steam and ultrasound caused a significant reduction in the b* parameter in the dried pomace.

## Figures and Tables

**Figure 1 molecules-26-03683-f001:**
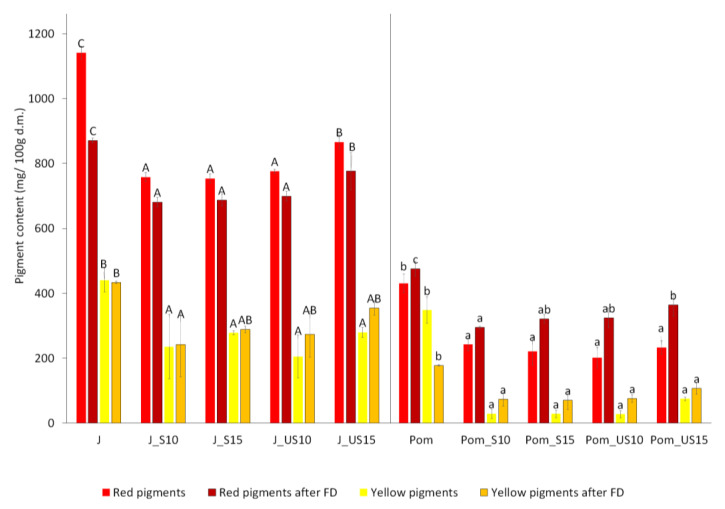
Pigments content. Sample abbreviation: J-Juice, Pom–Pomace, S-Steam, US–Ultrasound, 10, 15–time of pretreatment (min); A–D different capital letters in the each series for juices means statistical difference, a–d different small letters in each series for pomace means statistical difference.

**Table 1 molecules-26-03683-t001:** Physical properties of beetroot juice and pomace before and after freeze-drying.

	Pre-Treatment	Sample Name	Dry Matter (%)	Color Coefficients X_AV_ + SD			Dry Matter after FD (%) X_AV_ + SD	Color Coefficients after FD X_AV_ + SD		
			X_AV_ + SD	L*	a*	b*		L*	a*	b*
**Juice**	-	J	8.9 ± 0.8 ^A^	3.67 ± 0.31 ^A^	5.82 ± 0.15 ^A^	0.12 ± 0.08 ^A^	95.1 ± 0.7 ^C^	17.59 ± 0.51 ^B^	12.75 ± 0.48 ^C^	2.14 ± 0.23 ^D^
	S	J_S10	10.2 ± 0.4 ^AB^	3.97 ± 0.17 ^AB^	8.68 ± 0.32 ^B^	0.56 ± 033 ^B^	93.9 ± 09 ^AB^	15.65 ± 1.74 ^A^	7.32 ± 1.57 ^A^	0.13 ± 0.47 ^A^
		J_S15	10.5 ± 0.6 ^B^	4.01 ± 0.34 ^AB^	9.18 ± 0.23 ^B^	0.81 ± 0.18 ^B^	93.7 ± 0.3 ^A^	17.47 ± 0.84 ^B^	8.76 ± 1.32 ^AB^	0.61 ± 0.20 ^B^
	US	J_US10	10.3 ± 0.7 ^AB^	4.14 ± 0.25 ^B^	8.85 ± 0.94 ^B^	0.72 ± 0.45 ^B^	94.6 ± 0.4 ^BC^	17.44 ± 1.06 ^B^	12.41 ± 2.69 ^C^	1.47 ± 0.50 ^C^
		J_US15	9.5 ± 0.9 ^AB^	4.09 ± 0.24 ^B^	8.87 ± 1.09 ^B^	0.78 ± 0.42 ^B^	94.6 ± 0.1 ^A–C^	16.34 ± 0.29 ^AB^	10.60 ± 0.89 ^BC^	1.42 ± 0.21 ^C^
**Pomace**	-	Pom	17.9 ± 0.1 ^a^	10.15 ± 0.46 ^ab^	14.18 ± 0.31 ^ab^	2.55 ± 0.28 ^ab^	97.5 ± 0.0 ^ab^	27.80 ± 0.29 ^ab^	24.74 ± 0.13 ^b^	−0.30 ± 0.05 ^c^
	S	Pom_S10	18.4 ± 0.5 ^a^	10.68 ± 0.90 ^abc^	13.12 ± 1.67 ^a^	1.71 ± 0.71 ^a^	97.5 ± 0.1 ^ab^	26.13 ± 3.10 ^a^	22.96 ± 0.70 ^a^	−1.22 ± 0.19 ^b^
		Pom_S15	17.6 ± 0.1 ^a^	9.70 ± 0.65 ^a^	15.74 ± 0.73 ^b^	2.96 ± 0.31 ^b^	97.9 ± 0.4 ^b^	27.78 ± 0.52 ^ab^	23.77 ± 0.43 ^ab^	−1.10 ± 0.18 ^b^
	US	Pom_US10	18.1 ± 0.4 ^a^	11.02 ± 0.60 ^bc^	14.08 ± 0.94 ^a^	2.14 ± 0.52 ^a^	97.2 ± 0.3 ^a^	28.40 ± 0.95 ^b^	23.85 ± 0.27 ^b^	−2.10 ± 0.75 ^a^
		Pom_US15	18.3 ± 0.8 ^a^	11.30 ± 0.24 ^c^	15.62 ± 0.80 ^b^	2.97 ± 0.36 ^b^	97.9 ± 0.1 ^b^	27.14 ± 1.31 ^ab^	27.38 ± 1.05 ^c^	−1.12 ± 0.13 ^b^

Sample abbreviation: J–Juice, Pom–Pomace, S–Steam, US–Ultrasound; 10, 15–time of pretreatment (min); A–D different superscript, uppercase letters in the column for juices mean statistical difference, a–d different superscript, lowercase letters in column for pomace mean statistical difference.

## Data Availability

All data created and analyzed during the experiments were presented in this study.

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
