# Peer review of "The Influence of Different Pretreatment Methods on Color and Pigment Change in Beetroot Products"

_molecules, 2021, doi:10.3390/molecules26123683_

Round 1
Reviewer 1 Report
Minor remarks
Provide a blank space between quantity and unit.
Greek symbols and Latin names should be written in italics through the whole manuscript.
In the reference list, check the reference 20. Provide year, volume, issue, and pages.
Citation of references should be improved in the text. Suffice it to mention the name of only the first author.
Major remarks
The novelty of the manuscript is not well presented and should be clearly stated.
In the manuscript, avoid lumping the references. Each reference should be discussed separately.
All mentioned supplemented figures and tables should be avoided from the sentences. They should be presented in parentheses.
Why is the dry matter determined after the drying process at 75 °C? 105 °C is the recommended temperature for the determination of dry matter according to the standard procedure.
The resolution and size of the axis should be improved.
In this study, it is desirable to insert the results of spectrophotometric or chromatographic methods that indicate that there are not any significant structural changes in the obtained samples before and after treatment.
The conclusion should be improved and to give the main conclusions of this research.

Author Response
|
Dear Reviewer We would like to thank you for the review and all comments and suggestions. Blank spaces are now provided in the manuscript as well as the italic style to the Greek symbols and Latin names, we apologize for that mistake. Literature was made by EndNote program with Molecules style, we apologize for the mistakes. Now all citation in text and in references has been checked and corrected. Answers to major comments are listed below. |
|
|
Major remarks The novelty of the manuscript is not well presented and should be clearly stated. |
We have added new information to the article, mainly to the introduction and conclusion. We hope that now the novelty of the process in more visible. |
|
In the manuscript, avoid lumping the references. Each reference should be discussed separately. |
In introduction section some articles are gathered because in all of them the same or similar information are mentioned, and we decided to support the content presented in the article with several items from the literature. In discussion section, if data are discussed, we try to use one literature position for separate sentences. However, if, as in introduction section, we have mentioned information about process or properties of betalain pigments stated by different authors in some places we decided to leave multiple literature citation. In all multiple citation we try to use maximum 3 citations. |
|
All mentioned supplemented figures and tables should be avoided from the sentences. They should be presented in parentheses. |
It has been corrected. |
|
Why is the dry matter determined after the drying process at 75 °C? 105 °C is the recommended temperature for the determination of dry matter according to the standard procedure |
We have added more information in the text. “Dry matter was conducted in beetroot as well as in fresh and ground freeze-dried juices and pomace. For all samples gravimetrically method was used. About 0.6-1 g of sample was placed in a dish and dried by vacuum drying method (Memmert VO400, Schwabach, Germany) under the pressure of 10 mPa in 75 °C for 24 h until constant weight according to information from Rybak, et al. [46] and AOAC protocol [47]. Juices were placed on a dried filter paper placed in a dish, which prevents the juice from burning. Measurements were made in triplicate.”
In this protocol is mentioned that independently from material state we can used 75oC for 24 h.
What more we have decided to use this temperature for FD powders because of its structure which is delicate and hygroscopic, especially those from juice without carrier addition. Our experience shows that drying in 105oC can cause a structure “collapse” of such materials, which makes further drying difficult and, as a result, incorrect dry matter values are obtained. In 75oC evaporation is slower and no changes is observed. What more, in this method we have measured the weight at the end 2 times and mass did not changed in these conditions. |
|
The resolution and size of the axis should be improved. |
It has been corrected. |
|
In this study, it is desirable to insert the results of spectrophotometric or chromatographic methods that indicate that there are not any significant structural changes in the obtained samples before and after treatment. |
Thank you for that suggestion, in future experiments we will try to test structure by measuring the porosity of the sample before and after pretreatment and before pressing the juice. Now, unfortunately we do not have samples from this research, and new one could differ, because of the seasonality and storage of beetroot nowadays. Moreover, in presented article we would like to compare juice with pomaces obtained from beetroot treated by US and steam pretreatment methods not the beetroot itself, while the tests for beetroot slices are mentioned in some articles, but the juices and pomaces not. What more, in our opinion, testing the structure in pomaces which is destroyed by pressing process (in all experiments the same juicer was used) not influence on the pigment content and color, because of almost the same destruction during pressing process. |
|
The conclusion should be improved and to give the main conclusions of this research. |
It has been rewritten and more information is added. |
Reviewer 2 Report
Published research studies have established that freeze-drying processes can be effectively used on raw materials that are rich in carotenoids.
Some studies have shown that oven and natural convective drying causes greater loss of color than freeze-drying.
There are many advantages of freeze-drying, but attention should be paid to the limitations associated with this technique, including the long drying times and high operating costs.
The inherent instability of pigments is the main problem associated with natural colorants.
The authors have studied the effect of two pretreatments (thermal-steam and nonthermal-ultrasound) on juice and beetroot pomace.
Lyophilisation was chosen as drying method.
Physico-chemical (dry matter and color coefficients) and pigment content analyses were performed on all of the samples (juice and pomace before and after freeze drying with the two pretreatment options).
The results have been discussed in comparison with previous studies, indicating possible alterations of the compounds at high temperatures and under ultrasound exposure.
Revision
- The title needs to be rephrased. Numerous studies have already concluded that freeze-drying could be used to obtain pigments from various plant sources, including beetroots.
- Please rephrase and expand the conclusions section in correlation with the effects of the two pretreatments used for the juice and pomace.
- Please perform a language revision of the manuscript.
Conclusion
Accept after minor revision
Author Response
|
Dear Reviewer We would like to thank you for the review and all comments and suggestions. |
|
|
The title needs to be rephrased. Numerous studies have already concluded that freeze-drying could be used to obtain pigments from various plant sources, including beetroots. |
We decided to change title into: The influence of different pretreatment methods on color and pigment change in beetroot products |
|
Please rephrase and expand the conclusions section in correlation with the effects of the two pretreatments used for the juice and pomace. |
It has been rewritten and more information is added. |
|
Please perform a language revision of the manuscript. |
Manuscript was checked by Native English speaker. |
Round 2
Reviewer 1 Report
All minor remarks are given in the manuscript.
